# Building powerful and equivariant graph neural networks with structural message-passing

**Clément Vignac, Andreas Loukas, and Pascal Frossard**
EPFL
Lausanne, Switzerland
{clement.vignac,andreas.loukas,pascal.frossard}@epfl.ch

## Abstract

Message-passing has proved to be an effective way to design graph neural networks, as it is able to leverage both permutation equivariance and an inductive bias towards learning local structures in order to achieve good generalization. However, current message-passing architectures have a limited representation power and fail to learn basic topological properties of graphs. We address this problem and propose a powerful and equivariant message-passing framework based on two ideas: first, we propagate a one-hot encoding of the nodes, in addition to the features, in order to learn a *local context* matrix around each node. This matrix contains rich local information about both features and topology and can eventually be pooled to build node representations. Second, we propose methods for the parametrization of the message and update functions that ensure permutation equivariance. Having a representation that is independent of the specific choice of the one-hot encoding permits inductive reasoning and leads to better generalization properties. Experimentally, our model can predict various graph topological properties on synthetic data more accurately than previous methods and achieves state-of-the-art results on molecular graph regression on the ZINC dataset.

## 1 Introduction

Graph neural networks have recently emerged as a popular way to process and analyze graph-structured data. Among the numerous architectures that have been proposed, the class of message-passing neural networks (MPNNs) [1–3] has been by far the most widely adopted. In addition to being able to efficiently exploit the sparsity of graphs, MPNNs exhibit an inherent tendency to learn relationships between nearby nodes. This inductive bias is generally considered as a good fit for problems that require relational reasoning [4], such as tractable relational inference [5, 6], problems in combinatorial optimization [7–10] or the simulation of physical interactions between objects [11, 12].

A second key factor to the success of MPNNs is their equivariance properties. Since neural networks can ultimately only process tensors, in order to use a graph as input, it is necessary to order its nodes and build an adjacency list or matrix. Non-equivariant networks tend to exhibit poor sample efficiency as they need to explicitly learn that all representations of a graph in the (enormous) symmetry group of possible orderings actually correspond to the same object. On the contrary, permutation equivariant networks, such as MPNNs, are better equipped to generalize as they already implement the prior knowledge that any ordering is arbitrary.

Despite their success, equivariant MPNNs possess limited expressive power [13, 14]. For example, they cannot learn whether a graph is connected, what is the local clustering coefficient of a node, or if a given pattern such as a cycle is present in a graph [15]. For tasks where the graph structure is important, such as the prediction of chemical properties of molecules [16, 17] and the solution to combinatorial optimization problems, more powerful graph neural networks are necessary.

Aiming to address these challenges, this work puts forth *structural message-passing* (SMP)—a new type of graph neural network that is strictly more powerful than MPNNs, while also sharing the attractive inductive bias of message-passing architectures. SMP inherits its power from its ability to manipulate node identifiers. However, in contrast to previous studies that relied on identifiers [18, 19], it does so *in a permutation equivariant way* without introducing new sources of randomness. As a result, SMP can be powerful without sacrificing its ability to generalize to unseen data. In particular, we show that if SMP is built out of powerful layers, the resulting model is computationally universal over the space of equivariant functions.

Concretely, SMP maintains at each node a matrix called "local context" (instead of a feature vector as in MPNNs) that is initialized with a one-hot encoding of the nodes and the node features. These local contexts are then propagated in such a way that a permutation of the nodes or a change in the one-hot encoding will reorder the lines of each context without changing their content, which is key to efficient learning and good generalization.

We evaluate SMP on a diverse set of structural tasks that are known to be difficult for message-passing architectures, such as cycle detection, connectivity testing, diameter and shortest path distance computation. In all cases, our approach compares favorably to previous methods: for example, SMP solves cycle detection in all evaluated configurations, whereas other powerful networks struggle when the graphs become larger, and MPNNs do not manage to solve the task completely.

Finally, we evaluate our method on the ZINC chemistry dataset and achieve state-of-the-art performance among methods that do not use expert features. It shows that SMP is able to successfully learn both from the features and from topological information, which is essential in chemistry applications. Overall, our method is able to overcome a major limitation of MPNNs, while retaining their ability to process features with a bias towards locality.

**Notation.**    In the following, we consider the problem of representation learning on one or several graphs of possibly varying sizes. Each graph $G = (V, E)$ has an adjacency matrix $\boldsymbol{A} \in \mathbb{R}^{n \times n}$, and potentially node attributes $\boldsymbol{X} = (\boldsymbol{x}_1, ..., \boldsymbol{x}_n)^T \in \mathbb{R}^{n \times c_X}$ and edge attributes $\boldsymbol{y}_{ij} \in \mathbb{R}^{c_Y}$ for every $(v_i, v_j) \in E$. These attributes are aggregated into a 3-d tensor $\mathbf{Y} \in \mathbb{R}^{n \times n \times c_Y}$. We consider the edge weights of weighted graphs as edge attributes and view $\boldsymbol{A}$ as a binary adjacency matrix. The set of neighbors of a node $v_i \in V$ is written as $N_i$.

## 2   Related work

### 2.1   Permutation equivariant graph neural networks

Originally introduced by Scarselli et al. [1], MPPNs have progressively been extended to handle edge [2] and graph-level attributes [3]. Despite the flexibility in their parametrization, MPNNs without special node attributes however all have limited expressive power, even in the limit of infinite depth and width. For instance, they are at most as good at isomorphism testing as the Weisfeiler-Lehman (WL) vertex refinement algorithm [20, 13]. The WL test has higher dimensional counterparts ($k$-WL) of increasing power, which has motivated the introduction of the more powerful $k$-WL networks [14]. However, these higher-order networks are global, in the sense that they iteratively update the state of a $k$-tuple of nodes based on all other nodes (and not only neighbours), a procedure which is very costly both in time and memory. While a faster procedure was proposed in [21] concurrently to our work, key differences with SMP remain: we propose to learn richer embeddings for each node instead of one embedding per k-tuple of nodes, and build our theoretical analysis on distributed algorithms rather than vertex refinement methods.

Recent studies have also characterized the expressive power of MPNNs from other perspectives, such as the ability to approximate continuous functions on graphs [22] and solutions to combinatorial problems [23], highlighting similar limitations of MPNNs — see also  [24–28].

Beyond higher-order message-passing architectures, there have been efforts to construct more powerful equivariant networks. One way to do so is to incorporate hand-crafted topological features (such as the presence of cliques or cycles) [29], which requires expert knowledge on what features are relevant for a given task. A more task-agnostic alternative is to build networks by arranging together a set of simple permutation equivariant functions and operators. These building blocks are:

- Linear equivariant functions between tensors of arbitrary orders: a basis for these functions was computed by Maron et al. [30], by solving the linear system imposed by equivariance.
- Element-wise functions, applied independently to each feature of a tensor.
- Operators that preserve equivariance, such as $+$, $-$, tensor and elementwise products, composition and concatenation along the dimension of the channels.

Similarly to Morris et al. [14], networks built this way obtain a better expressive power than MPNN by using higher-order tensors [31, 30]. Since $k$-th order tensors can represent any $k$-tuple of nodes, architectures manipulating them can exploit more information to compute topological properties (and be as powerful as the $k$-WL test). Unfortunately, memory requirements are exponential in the tensor order, which makes these methods of little practical interest. More recently, Maron et al. [32] proposed provably powerful graph networks (PPGN) based on the observation that the use of matrix multiplication can make their model more expressive for the same tensor order. This principle was also used in the design of Ring-GNN [22], which has many similarities with PPGN. Key differences between such methods and ours are that (*i*) SMP can be parametrized to have a lower time complexity, due to the ability of message-passing to exploit the sparsity of adjacency matrices, (*ii*) SMP retains the message-passing inductive bias, which is different from PPGN and, as we will show empirically, makes it better suited to practical tasks such as the detection of substructures in a graph.

## 2.2 Non-equivariant graph neural networks

In order to better understand the limitations of current graph neural networks, analogies with graph theory and distributed systems have been exploited. In these fields, a large class of problems cannot be solved without using node identifiers [33, 34]. The reasoning is that, in message-passing architectures, each node has access to a local view of the graph created by the reception of messages. Without identifiers, each node can count the number of incoming messages and process them, but cannot tell from how many unique nodes they come from. They are therefore unable to reconstruct the graph structure.

This observation has motivated researchers to provide nodes with randomly selected identifiers [18, 19, 35, 36]. Encouragingly, by showing the equivalence between message-passing and a model in distributed algorithms, Loukas [19] proved that graph neural networks with identifiers and sufficiently expressive message and update functions can be Turing universal, which was also confirmed on small instances of the graph isomorphism problem [37].

Nevertheless, the main issue with these approaches is sample efficiency. Identifiers introduce a dependency of the network on a random input and the loss of permutation equivariance, causing poor generalization. Although empirical evidence has been presented that the aforementioned dependency can be overcome with large amounts of training data or other augmentations [18, 37], overfitting and optimization issues can occur. In this work, we propose to overcome this problem by introducing a network which is both powerful and permutation equivariant.

## 3 Structural message-passing

We present the structural message-passing neural networks (SMP), as generalization of MPNNs that follows a similar design principle. However, rather than processing vectors with permutation invariant operators, SMP propagates *matrices* and processes them in a permutation equivariant way. This subtle change greatly improves the network's ability of to learn information about the graph structure.

### 3.1 Method

In SMP, each node of a graph maintains a *local context* matrix $\boldsymbol{U}_i \in \mathbb{R}^{n \times c}$ rather than a feature vector $\boldsymbol{x}_i \in \mathbb{R}^c$ as in MPNN. The $j$-th row of $\boldsymbol{U}_i$ contains the $c$-dimensional representation that node $v_i$ has of node $v_j$. Intuitively, equivariance means that the lines of the local context after each layer are simply permuted when the nodes are reordered, as shown in Fig. 1.

**Initialization** The local context is initialized as a one-hot encoding $\boldsymbol{U}_i^{(0)} = \mathbb{1}_i \in \mathbb{R}^{n \times 1}$ for every $v_i \in V$, which corresponds to having initially a unique identifier for each node. In addition, if there are features $\boldsymbol{x}_i$ associated with node $v_i$, they are appended to the same row of the local context as the

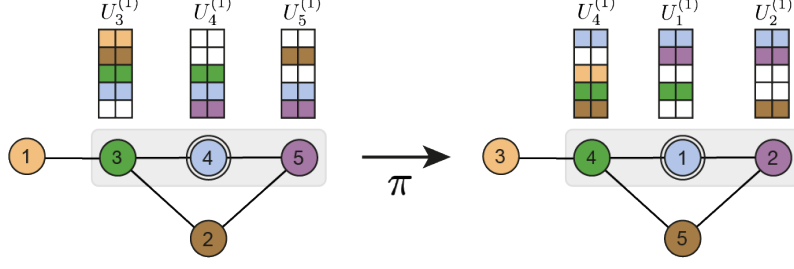

Figure 1: In the SMP model, each local context $\boldsymbol{U}_i^{(l)}$ is an $n \times c_l$ matrix, with each row storing the $c_l$-dimensional representation of a node (denoted by color). The figure shows the local context in the output of the first layer and blank rows correspond to nodes that have not been encountered yet. Upon node reordering, the lines of the local context are permuted but their content remains unchanged.

identifiers: $\boldsymbol{U}_i^{(0)}[i, :] = [1, \boldsymbol{x}_i] \in \mathbb{R}^{1+c_X}$. Later, we will show that when SMP is parametrized in the proper way, the ordering induced by the one-hot encoding is actually irrelevant to the output.

**Layers** At layer $l + 1$, the state of each node is updated as in standard MPNNs [3]: messages are computed on each edge before being aggregated into a single matrix via a symmetric function. The result can then be updated using the local context of previous layer at this node:

$$\boldsymbol{U}_i^{(l+1)} = u^{(l)}\left(\boldsymbol{U}_i^{(l)}, \tilde{\boldsymbol{U}}_i^{(l)}\right) \in \mathbb{R}^{n \times c_{l+1}} \quad \text{with} \quad \tilde{\boldsymbol{U}}_i^{(l)} = \phi\left(\left\{m^{(l)}(\boldsymbol{U}_i^{(l)}, \boldsymbol{U}_j^{(l)}, \boldsymbol{y}_{ij})\right\}_{v_j \in N_i}\right)$$

Above, $u^{(l)}$, $m^{(l)}$, $\phi$ are the *update*, *message* and *aggregation* functions of the $(l + 1)$-th layer, respectively, whereas $c_{l+1}$ denotes the layer's width.

It might be interesting to observe that, starting from a one-hot encoding and using the update rule $\boldsymbol{U}_i^{(l+1)} = \sum_{v_j \in N_i} \boldsymbol{U}_j^{(l)}$, SMP iteratively compute powers of $\boldsymbol{A}$. Since $\boldsymbol{A}^l[i, j]$ corresponds to the count of walks of length $l$ between $v_i$ and $v_j$, there is a natural connection between the propagation of identifiers and the detection of topological features: even with simple parametrizations, SMP can manipulate polynomials in the adjacency matrix and therefore learn spectral properties [38] that MPNNs cannot detect.

In the following, it will be convenient to express each SMP layer $f^{(l)}$ in a tensor form:

$$\mathbf{U}^{(l+1)} = f^{(l)}(\mathbf{U}^{(l)}, \mathbf{Y}, \boldsymbol{A}) = [\boldsymbol{U}_1^{(l+1)}, \ldots, \boldsymbol{U}_n^{(l+1)}] \in \mathbb{R}^{n \times n \times c_{l+1}}$$

**Pooling** After all $L$ message-passing layers have been applied, the aggregated contexts $\mathbf{U}^{(L)}$ can be pooled to a vector or to a matrix (e.g, for graph and node classification, respectively). To obtain an equivariant representation for node classification, we aggregate each $\boldsymbol{U}_i^{(L)} \in \mathbb{R}^{n \times c_L}$ into a vector using an equivariant neural network for sets $\sigma$ [39–42] applied simultaneously to each node $v_i$:

$$f_{eq}(\mathbf{U}^{(0)}, \mathbf{Y}, \boldsymbol{A}) = \sigma \circ f^{(L)} \circ \cdots \circ f^{(1)}(\mathbf{U}^{(0)}, \mathbf{Y}, \boldsymbol{A}) \in \mathbb{R}^{n \times c},$$

whereas a permutation invariant representation is obtained after the application of a pooling function *pool*. It may be a simple sum or average followed by a soft-max, or a more complex operator [43]:

$$f_{inv}(\mathbf{U}^{(0)}, \mathbf{Y}, \boldsymbol{A}) = pool \circ f_{eq}(\mathbf{U}^{(0)}, \mathbf{Y}, \boldsymbol{A}) \in \mathbb{R}^c$$

## 3.2 Analysis

The following section characterizes the equivariance properties and representation power of SMP. For the sake of clarity, we defer all proofs to the appendix.

**Equivariance** Before providing sufficient conditions for permutation equivariance, we define it formally. A change in the ordering of $n$ nodes can be described by a permutation $\pi$ of the symmetric group $\mathfrak{S}_n$. $\pi$ acts on a tensor by permuting the axes indexing nodes (but not the other axes):

$$(\pi \, . \, \mathbf{U})[i, j, k] = \mathbf{U}[\pi^{-1}(i), \, \pi^{-1}(j), k], \quad \text{where} \quad \mathbf{U} \in \mathbb{R}^{n \times n \times c}$$

For vector and matrices, the action of a permutation is more easily described using the matrix $\mathbf{\Pi}$ canonically associated to $\pi$: $\pi.z = z$ for $z \in \mathbb{R}^c$, $\pi.\boldsymbol{X} = \mathbf{\Pi}^T \boldsymbol{X}$ for $\boldsymbol{X} \in \mathbb{R}^{n \times c}$, and $\pi.\boldsymbol{A} = \mathbf{\Pi}^T \boldsymbol{A} \mathbf{\Pi}$ for $\boldsymbol{A} \in \mathbb{R}^{n \times n}$. An SMP layer $f$ is said to be permutation equivariant if permuting the inputs and applying $f$ is equivalent to first applying $f$ and then permuting the result:

$$\forall \pi \in \mathfrak{S}_n, \quad \pi \,.\, f(\mathbf{U}, \mathbf{Y}, \boldsymbol{A}) = f(\pi.\mathbf{U}, \pi.\mathbf{Y}, \pi.\boldsymbol{A}))$$

We stress that an equivariant SMP network should yield the same output (up to a permutation) for every one-hot encoding used to construct the node identifiers. We can now state some sufficient conditions for equivariance:

**Theorem 1** (Permutation equivariance). *Let functions $m$, $\phi$ and $u$ be permutation equivariant, that is, for every permutation $\pi \in \mathfrak{S}_n$ we have $u(\pi.\boldsymbol{U}, \pi.\boldsymbol{U}') = \pi.u(\boldsymbol{U}, \boldsymbol{U}')$, $\phi(\{\pi.\boldsymbol{U}_j\}_{v_j \in N_i}) = \pi.\phi(\{\boldsymbol{U}_j\}_{v_j \in N_i})$, and $m(\pi.\boldsymbol{U}, \pi.\boldsymbol{U}', \boldsymbol{y}) = \pi.m(\boldsymbol{U}, \boldsymbol{U}', \boldsymbol{y})$. Then, SMP is permutation equivariant.*

The proof is presented in Appendix A. This theorem defines the class of functions that can be used in our model. For example, if the message and update functions are operators applied simultaneously to each row of the local context, the whole layer is guaranteed to be equivariant. However, more general functions can be used: each $\boldsymbol{U}_i$ is a $n \times c$ matrix which can be viewed as the representation of a set of nodes. Hence, any equivariant neural network for sets can be used, which allows the network to have several desirable properties:

- *Inductivity*: as an equivariant neural network for sets can take sets of different size as input, SMP can be trained on graphs with various sizes as well. Furthermore, it can be used in inductive settings on graphs whose size has not been seen during training, which we will confirm experimentally.
- *Invariance to local isomorphisms*: SMP learns *structural embeddings*, in the sense that it yields the same result on isomorphic subgraphs. More precisely, if the subgraphs $G_i^k$ and $G_j^k$ induced by $G$ on the k-hop neighborhoods of $v_i$ and $v_j$ are isomorphic, then on node classification, any $k$-layer SMP $f$ will yield the same result for $v_i$ and $v_j$. This is in contrast with several popular methods [44, 45] that learn *positional embeddings* which do not have this property.

**Representation and expressive power**   The following theorem characterizes the representation power of SMP when parametrized with powerful layers. Simply put, Theorem 2 asserts that it is possible to parameterize an SMP network such that it maps non-isomorphic graphs to different representations:

**Theorem 2** (Representation power – informal). *Consider the class $S$ of simple graphs with $n$ nodes, diameter at most $\Delta$ and degree at most $d_{max}$. We assume that these graphs have respectively $c_X$ and $c_Y$ attributes on the nodes and the edges. Then, there exists a SMP network $f$ of depth at most $\Delta + 1$ and width at most $2d_{max} + c_X + n\, c_Y$ such that the full structure of any graph in $S$ (with the attributes) can be recovered from the output of $f$ at any node.*

The formal statement and the proof are detailed in Appendix B. We first show the result for the simple case where $f$ can pass messages of size $n \times n$, and then consider the case of $n \times 2d_{\max}$ matrices using the following lemma:

**Lemma 1** (Maehara and Rödl [46]). *For any simple graph $G = (V, E)$ of $n$ nodes and maximum degree $d_{max}$, there exists a unit-norm embedding of the nodes $\boldsymbol{X} \in \mathbb{R}^{n \times 2d_{max}}$ such that for every $v_i, v_j \in V$, $(v_i, v_j) \in E \iff \langle \boldsymbol{X}_i, \boldsymbol{X}_j \rangle = 0$.*

The universality of SMP is a direct corollary: since each node can have the ability to reconstruct the adjacency matrix from its local context, it can also employ a universal network for sets [39] to compute any equivariant function on the graph (cf. Appendix C). Interestingly, this result shows that propagating matrices instead of vectors might be a way to solve the bottleneck problem [47]: while MPNNs need feature maps that exponentially grow with the graph size in order to recover the topology, SMPs can do it with $O(d_{\max} n^2)$ memory.

**Corollary 1** (Expressive power). *Let $G$ be a simple graph of diameter at most $\Delta$ and degree at most $d_{max}$. Consider an SMP $f = f^{(L)} \circ \cdots \circ f^{(1)}$ of depth $L = \Delta$ and width $2d_{max} + c_X + n\, c_Y$ satisfying the properties of Theorem 3. Then, any equivariant function can be computed as $f_{eq} = \sigma \circ f$, where $\sigma$ is a universal function of sets applied simultaneously to each node. Similarly, any permutation invariant function can be computed as $f_{in} = \frac{1}{n} \sum_{v_i \in V} \sigma \circ f$.*

These results show that two components are required to build a universal approximator of functions on graphs. First, one needs an algorithm that breaks symmetry during message passing, which SMP manages to do in an equivariant manner. Second, one needs powerful layers to parametrize the message, aggregation and update functions. Here, we note that the proofs of Theorem 2 and Corollary 1 are not constructive and that deriving practical parametrizations that are universal remains an open question [48]. Nevertheless, we do constructively prove the following more straightforward claim using a simple parametrization:

**Proposition 1.** *SMP is strictly more powerful than MPNN: SMP can simulate any MPNN with the same number of layers, but MPNNs cannot simulate all SMPs.*

To prove it, we create for any MPNN a corresponding SMP which performs the same operations as the MPNN on the main diagonal of the local context. On the contrary, we can easily create an SMP which is able to distinguish between two small graphs that cannot be distinguished by the Weisfeiler-Lehman test (Appendix D).

## 4 Implementation

SMP offers a lot of flexibility in its implementation, as any equivariant function that combines the local context of two nodes and the edge features can be used. We propose two implementations that we found to work well, but our framework can also be implemented differently. In both cases, we split the computation of the messages in two steps. First, the local context of each node is updated using a neural network for sets. Then, a standard message passing network is applied separately on each row of the local contexts. For the first step, we use a subset of the linear equivariant functions computed by Maron et al. [30]:

$$\forall v_i \in V, \quad \hat{\boldsymbol{U}}_i^{(l)} = \boldsymbol{U}_i^{(l)} \boldsymbol{W}_1^{(l)} + \frac{1}{n} \mathbf{1}_n \mathbf{1}_n^T \boldsymbol{U}_i^{(l)} \boldsymbol{W}_2^{(l)} + \mathbf{1}_n (\boldsymbol{c}^{(l)})^\top + \frac{1}{n} \mathbb{1}_i \mathbf{1}^T \boldsymbol{U}_i^{(l)} \boldsymbol{W}_3^{(l)},$$

where $\mathbf{1}_n \in \mathbb{R}^{n \times 1}$ is a vector of ones, $\mathbb{1}_i \in \mathbb{R}^{n \times 1}$ the indicator of $v_i$, whereas $(\boldsymbol{W}_k^{(l)})_{1 \leq k \leq 5}$ and $\boldsymbol{c}^{(l)}$ are learnable parameters. As for the message passing architecture, we propose two implementations with different computational complexities:

**Default SMP** This architecture corresponds to a standard MPNN, where the message and update functions are two-layer perceptrons. We use a sum aggregator normalized by the average degree $d_{\text{avg}}$ over the graph: it retains the good properties of the sum aggregator [13], while also avoiding the exploding-norm problem [49]. This network can be written:

$$m_{\text{def}}^{(l)}(\hat{\boldsymbol{U}}_i^{(l)}, \hat{\boldsymbol{U}}_j^{(l)}, \boldsymbol{y}_{ij}) = MLP(\hat{\boldsymbol{U}}_i^{(l)}, \hat{\boldsymbol{U}}_j^{(l)}, \boldsymbol{y}_{ij}) \tag{1}$$
$$\boldsymbol{U}_i^{(l+1)} = MLP(\hat{\boldsymbol{U}}_i^{(l)}, \textstyle\sum_{v_j \in N_i} m_{\text{def}}^{(l)}(\boldsymbol{U}_i^{(l)}, \boldsymbol{U}_j^{(l)}, \boldsymbol{y}_{ij})/d_{\text{avg}}),$$

**Fast SMP** For graphs without edge features, we propose a second implementation with a message function that uses a pointwise multiplication $\odot$:

$$m_{\text{fast}}^{(l)}(\hat{\boldsymbol{U}}_i^{(l)}, \hat{\boldsymbol{U}}_j^{(l)}) = \hat{\boldsymbol{U}}_j^{(l)} + \left(\hat{\boldsymbol{U}}_i^{(l)} \boldsymbol{W}_4^{(l)}\right) \odot \left(\hat{\boldsymbol{U}}_j^{(l)} \boldsymbol{W}_5^{(l)}\right),$$

where $\boldsymbol{W}_4$ and $\boldsymbol{W}_5$ are learnable matrices. The aggregation is the same, and the update is simply a residual connection, so that the $l$-th SMP layer updates each node's local context as

$$\boldsymbol{U}_i^{(l+1)} = \hat{\boldsymbol{U}}_i^{(l)} + \frac{1}{d_{\text{avg}}} \textstyle\sum_{v_j \in N_i} m_{\text{fast}}^{(l)}(\boldsymbol{U}_i^{(l)}, \boldsymbol{U}_j^{(l)})$$
$$= \hat{\boldsymbol{U}}_i^{(l)} + \left(\textstyle\sum_{v_j \in N_i} \hat{\boldsymbol{U}}_j^{(l)} + \hat{\boldsymbol{U}}_i^{(l)} \boldsymbol{W}_4^{(l)} \odot \sum_{v_j \in N_i} \hat{\boldsymbol{U}}_j^{(l)} \boldsymbol{W}_5^{(l)}\right)/d_{\text{avg}}$$

In this last equation, the arguments of the two sums are only functions of the local context of node $v_j$. This allows for a more efficient implementation, where one message is computed per node, instead of one per edge as in default SMP.

One might notice that Fast SMP can be seen as a local version of PPGN (proof in Appendix F):

**Proposition 2.** *A Fast SMP with $k$ layers can be approximated by a $2k$-block PPGN.*

Despite not being more powerful, Fast SMP has the advantage of being more efficient than PPGN, as it can exploit the sparsity of adjacency matrices. Furthermore, as we will see experimentally, our method manages to learn topological information much more easily than PPGN, a property that we attribute to the inductive bias carried by message-passing.

Table 1: Time and space complexity of the forward pass expressed in terms of number of nodes $n$, number of edges $m$, number of node colors $\chi$, and width $c$. For connected graphs, we trivially have $\chi \leq n \leq m + 1 \leq n^2$.

| Method | Memory per layer | Time complexity per layer |
|---|---|---|
| GIN [13] | $\Theta(n\,c)$ | $\Theta(m\,c + n\,c^2)$ |
| MPNN [2] | $\Theta(n\,c)$ | $\Theta(m\,c^2)$ |
| Fast SMP (with coloring) | $\Theta(n\,\chi\,c)$ | $\Theta(m\,\chi\,c + n\,\chi\,c^2)$ |
| Fast SMP | $\Theta(n^2\,c)$ | $\Theta(m\,n\,c + n^2\,c^2)$ |
| SMP | $\Theta(n^2\,c)$ | $\Theta(m\,n\,c^2)$ |
| PPGN [32] | $\Theta(n^2\,c)$ | $\Theta(n^3\,c + n^2\,c^2)$ |
| Local order-3 WL [14] | $\Theta(n^3\,c)$ | $\Theta(n^4\,c + n^3\,c^2)$ |

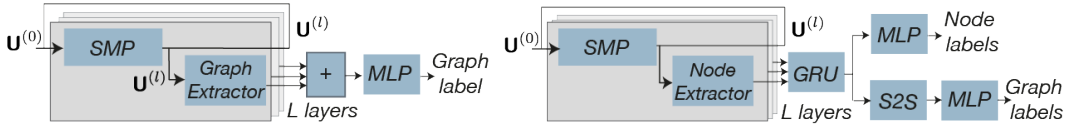

Figure 2: (left) Architecture for cycle detection. The graph extractor computes the trace and the sum along the two first axes of $\mathbf{U}$, and passes the result into a two-layer MLP in order to produce a set of global features. (right) Architecture for multi-task learning: after each convolution, node features are extracted using a two-layer MLP followed by three pooling methods (mean, max, and the extraction of $\mathbf{U}[i, i, :]$ for each $v_i \in V$), and a final linear layer. The rest of the architecture is similar to Corso et al. [50]: it uses a Gated Recurrent Unit (GRU) and a Set-to-set network (S2S).

**Complexity**    Table 1 compares the per-layer space and time complexity induced by the forward pass of SMP with that of other standard graph networks. Whereas local order-3 Weisfeiler-Lehman networks need to store all triplets of nodes, both PPGN and SMP only store information for pairs of nodes. However, message-passing architectures (such as SMP) can leverage the sparsity of the adjacency matrix and hence benefit from a more favorable time complexity than architectures which perform global updates (as PPGN).

An apparent drawback of SMP (shared by all equivariant powerful architectures we are aware of) is the need for more memory than MPNN. This difference is partially misleading since it is known that the width of any MPNN needs to grow at least linearly with $n$ (for any constant depth) for it to be able to solve many graph-theoretic problems [19, 50, 37]. However, for graphs with a large diameter, the memory requirements of SMP can be relaxed by using the following observation: *if each node is colored differently from all nodes in its $2k$-hop neighborhood, then no node will see the same color twice in its $k$-hop neighborhood.* It implies that nodes which are far apart can use the same identifier without conflict. We propose in Appendix E a procedure (Fast SMP with coloring) based on greedy coloring which can replace the initial one-hot encoding, so that each node can manipulate smaller matrices $\boldsymbol{U}_i$. This method allows to theoretically improve both the time and space complexity of SMP, although the number of colors needed usually grows fast with the number of layers in the network.

## 5    Experiments

### 5.1    Cycle detection

We first evaluate different architectures on the detection of cycles of length 4, 6 and 8 (for several graph sizes), implemented as a graph classification problem[1]. Models are retrained for each cycle length and graph size on 10k samples with balanced classes, and evaluated on $10,000$ samples as well. The same architecture (detailed in Figure 2) is used for all models, as we found it to perform better than the original implementation of each method: the methods under comparison thus only differ in the definition of the convolution, making comparison easy. We use the fast implementation of SMP, as we find its expressivity to be sufficient for this task.

Table 2: Experiments on cycle detection, viewed as a graph classification problem.

(a) Test accuracy on the detection of cycles of various length with 10,000 training samples. (Best seen in color.) Only SMP solves the problem in all configurations.

| Cycle length | 4 | | | | 6 | | | | 8 | | | |
|---|---|---|---|---|---|---|---|---|---|---|---|---|
| Graph size | 12 | 20 | 28 | 36 | 20 | 31 | 42 | 56 | 28 | 50 | 66 | 72 |
| MPNN | 98.5 | 93.2 | 91.8 | 86.7 | 98.7 | 95.5 | 92.9 | 88.0 | 98.0 | 96.3 | 92.5 | 89.1 |
| GIN | 98.3 | 97.1 | 95.0 | 93.0 | 99.5 | 97.2 | 95.1 | 92.7 | 98.5 | 98.8 | 90.8 | 92.5 |
| GIN + degree | 99.3 | 98.2 | 97.3 | 96.7 | 99.2 | 97.1 | 97.1 | 94.5 | 99.3 | 98.7 | | 95.4 |
| GIN + rand id | 99.0 | 96.2 | 94.9 | 88.3 | 99.0 | 97.8 | 95.1 | 96.1 | 98.6 | 98.0 | 97.2 | 95.3 |
| RP [18] | 100 | 99.9 | 99.7 | 97.7 | 99.0 | 97.4 | 92.1 | 84.1 | 99.2 | 97.1 | 92.8 | 80.6 |
| PPGN | 100 | 100 | 100 | 99.8 | 98.3 | 99.4 | 93.8 | 87.1 | 99.9 | 98.7 | 84.4 | 76.5 |
| Ring-GNN | 100 | 99.9 | 99.9 | 99.9 | 100 | 100 | 100 | 100 | 99.1 | 99.8 | 74.4 | 71.4 |
| SMP | 100 | 100 | 100 | 100 | 100 | 100 | 100 | 100 | 100 | 100 | 100 | 99.9 |

(b) Test accuracy (%) when evaluating the generalization ability of inductive networks. Each network is trained on one graph size ("In-distribution"), validated on a second size, then tested on a third ("Out-of-distribution"). SMP is the only powerful network evaluated that generalizes well. *OOM* = out of memory.

| Setting | In-distribution | | | Out-of-distribution | | |
|---|---|---|---|---|---|---|
| Cycle length | 4 | 6 | 8 | 4 | 6 | 8 |
| Graph size | 20 | 31 | 50 | 36 | 56 | 72 |
| GIN | 93.9 | 99.7 | 98.8 | 81.1 | 85.8 | **88.8** |
| PPGN | 99.9 | 99.5 | 98.7 | 50.0 | 50.0 | 50.0 |
| Ring-GNN | 100 | 100 | 99.9 | 50.0 | 50.0 | *OOM* |
| SMP | 100 | 99.8 | 99.5 | **99.8** | **87.8** | 79.5 |

(c) Test accuracy (%) on the detection of 6 cycles for graphs with 56 nodes trained on less data. Thanks to its equivariance properties, SMP requires much less data for training.

| Train samples | 200 | 500 | 1000 | 5000 |
|---|---|---|---|---|
| GIN + random identifiers | 65.8 | 70.8 | 80.6 | 96.4 |
| SMP | **87.7** | **97.4** | **97.6** | **99.5** |

Results are shown in Tab 2. For a given cycle length, the task becomes harder as the number of nodes in the graph grows: the bigger the graph, the more candidate paths that the network needs to verify as being cycles. SMP is able to solve the task almost perfectly for all graph and cycle sizes. For standard message-passing models, we observe a correlation between accuracy and the presence of identifiers: random identifiers and weak identifiers (a one-hot encoding of the degree) tend to perform better than the baseline GIN and MPNN. PPGN and RING-GNN solve the task well for small graphs, but fail when $n$ grows. Perhaps due to a miss-aligned inductive bias, we encountered difficulties with training them, whereas message-passing architectures could be trained more easily. We provide a more detailed comparison between SMP, PPGN and Ring-GNN in Appendix G. We also compare the generalization ability of the different networks that can be used in inductive settings. GIN generalizes well, but SMP is the only one that achieves good performance among the powerful networks. This may be imputable to the inductive bias of message passing architectures, shared by GIN and SMP. Finally, we compare SMP and GIN with random identifiers in settings with less training data: SMP requires much fewer samples to achieve good performance, which confirms that equivariance is important for good sample efficiency.

## 5.2 Multi-task detection of graph properties

We further benchmark SMP on the multi-task detection of graph properties proposed in Corso et al. [50]. The goal is to estimate three node-defined targets: *geodesic distance* from a given node (Dist.), *node eccentricity* (Ecc.), and computation of *Laplacian features $Lx$* given a vector $x$ (Lap.), as well as three graph-defined targets: *connectivity* (Conn.), *graph diameter* (Diam.), and *spectral radius* (Rad.). The training set is composed of 5120 graphs with up to 24 nodes, while graphs in the test set have up to 19 nodes. Several MPNNs are evaluated as well as PNA [50], a message-passing model

Table 3: Log MSE on the test set (lower is better). Baseline results are from Corso et al. [50].

| Model | *Average* | Dist. | Ecc. | Lap. | Conn. | Diam. | Rad. |
|---|---|---|---|---|---|---|---|
| GIN | $-1.99$ | $-2.00$ | $-1.90$ | $-1.60$ | $-1.61$ | $-2.17$ | $-2.66$ |
| GAT | $-2.26$ | $-2.34$ | $-2.09$ | $-1.60$ | $-2.44$ | $-2.40$ | $-2.70$ |
| MPNN (sum) | $-2.53$ | $-2.36$ | $-2.16$ | $-2.59$ | $-2.54$ | $-2.67$ | $-2.87$ |
| PNA | $-3.13$ | $-2.89$ | $-2.89$ | $-3.77$ | $-2.61$ | $-3.04$ | $-3.57$ |
| Fast MPNN (Ablation) | $-2.37$ | $-2.47$ | $-1.99$ | $-2.83$ | $-1.61$ | $-2.40$ | $-2.93$ |
| MPNN (Ablation) | $-2.77$ | $-3.18$ | $-2.05$ | $-3.27$ | $-2.24$ | $-2.88$ | $-2.97$ |
| **Fast SMP** | $-3.53$ | $-3.31$ | $-3.36$ | $\mathbf{-4.30}$ | $-2.72$ | $\mathbf{-3.65}$ | $\mathbf{-3.82}$ |
| **SMP** | $\mathbf{-3.59}$ | $\mathbf{-3.59}$ | $\mathbf{-3.67}$ | $-4.27$ | $\mathbf{-2.97}$ | $-3.58$ | $-3.46$ |

Table 4: Mean absolute error (MAE) on ZINC, trained on a subset of 10k molecules.

| Model | No edge features | With edge features |
|---|---|---|
| Gated-GCN [53] | 0.435 | 0.282 |
| GIN [53] | 0.408 | 0.252 |
| PNA [50] | 0.320 | 0.188 |
| DGN [54] | **0.219** | 0.168 |
| MPNN-JT [53] | – | 0.151 |
| MPNN (ablation) | 0.272 | 0.189 |
| **SMP (Ours)** | **0.219** | **0.138** |

based on the combination of several aggregators. Importantly, random identifiers are used for all these models, so that all baseline methods are theoretically poweful [19], but not equivariant.

All models are benchmarked using the same architecture, apart from the fact that SMP manipulates local contexts. In order to pool these contexts into node features and use them as input to the Gated Recurrent Unit [52], we use an extractor described in Figure 2. As an ablation study, we also consider for each model a corresponding MPNN with the same architecture.

The results are summarized in Table 3. We find that both SMPs are able to exploit the local contexts, as they perform much better than the corresponding MPNN. SMP also outperforms other methods by a significant margin. Lastly, standard SMP tends to achieve better results than fast SMP on tasks that require graph traversals (shortest path computations, excentricity, checking connectivity), which may be due to a better representation power.

### 5.3 Constrained solubility regression on ZINC

The ZINC database is a large scale dataset containing molecules with up to 37 atoms. The task is to predict the constrained solubility of each molecule, which can be seen as a graph regression problem. We follow the setting of [55]: we train SMP on the same subset of 10,000 molecules with a parameter budget of around 100k, reduce the learning rate when validation loss stagnates, and stop training when it reaches a predefined value. We use an expressive parametrization of SMP, with 12 layers and 2-layer MLPs both in the message and the update functions. In order to reduce the number of parameters, we share the same feature extractor after each layer (cf Fig. 2). Results are presented in Table 4. They show that in both cases (with or without edge features, which are a one-hot encoding of the bond type), SMP is able to achieve state of the art performance. Note however than even better results (0.108 MAE using a MPNN with edge features [29]) can be achieved by augmenting the input with expert features. We did not use them in order to compare fairly with the baseline results.

## 6 Conclusion

We introduced structural message-passing (SMP), a new architecture that is both powerful and permutation equivariant, solving a major weakness of previous message-passing networks. Empirically, SMP significantly outperforms previous models in learning graph topological properties, but retains the inductive bias of MPNNs and their good ability to process node features. We believe that our work paves the way to graph neural networks that efficiently manipulate both node and topological features, with potential applications to chemistry, computational biology and neural algorithmic reasoning.

## Broader Impact

This paper introduced a new methodology for building graph neural networks, conceived independently of a specific application. As graphs constitute a very abstract way to represent data, they have found a lot of different applications [56]. The wide applicability of graph neural networks makes it challenging to foresee how our method will be used and the ethical problems which might occur.

Nevertheless, as we propose to overcome limitations of previous work in learning topological information, our method is likely to be used first and foremost in fields were graph topology is believed to be important. We hope in particular that it can contribute to the fields of quantum chemistry and drug discovery. The good performance obtained on the ZINC dataset is an encouraging sign of the potential of SMP in these fields. Other applications come to mind: material science [57], computational biology [58], combinatorial optimization [7–9] or code generation [59].

## Acknowledgments and Disclosure of Funding

Clément Vignac would like to thank the Swiss Data Science Center for supporting him through the PhD fellowship program (grant P18-11). Andreas Loukas would like to thank the Swiss National Science Foundation for supporting him in the context of the project "Deep Learning for Graph-Structured Data" (grant number PZ00P2 179981).

## Footnotes

[1]Our implentation with Pytorch Geometric [51] is available at `github.com/cvignac/SMP`.

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
