[Supplementary Material]

# A    Proof of Theorem 1

Let $f$ be a layer of SMP:

$$f(\mathbf{U}, \mathbf{Y}, \boldsymbol{A})[i, :, :] = u(\boldsymbol{U}_i, \phi(\{m(\boldsymbol{U}_i, \boldsymbol{U}_j, \boldsymbol{y}_{ij})\}_{v_j \in N_i})) = u(\boldsymbol{U}_i, \phi(\{m(\boldsymbol{U}_i, \boldsymbol{U}_j, \boldsymbol{y}_{ij})\}_{v_j : A[i,j]>0}))$$

The action of a permutation $\pi$ on the inputs is defined as $f(\pi.(\mathbf{U}, \mathbf{Y}, \boldsymbol{A})) = f(\pi.\mathbf{U}, \pi.\mathbf{Y}, \pi.\boldsymbol{A})$. In order to simplify notation, we will consider $\pi^{-1}$ instead of $\pi$. We have for example $(\pi^{-1}.\boldsymbol{A})[i, j] = A[\pi_i, \pi_j]$ and $(\pi^{-1}.\mathbf{U})[i, j, k] = \mathbf{U}[\pi_i, \pi_j, k]$, which can be written as

$$(\pi^{-1}.\mathbf{U})[i, :, :] = \boldsymbol{\Pi}\, \boldsymbol{U}_{\pi_i}.$$

As shown next, the theorem's conditions suffice to render SMP equivariant:

$$
\begin{aligned}
f(\pi^{-1}.(\mathbf{U}, \mathbf{Y}, \boldsymbol{A}))[i, :, :] &= u(\boldsymbol{\Pi}\, \boldsymbol{U}_{\pi_i},\ \phi(\{m(\boldsymbol{\Pi}\, \boldsymbol{U}_{\pi_i},\ \boldsymbol{\Pi}\, \boldsymbol{U}_{\pi_j},\ \boldsymbol{y}_{\pi_i \pi_j})\}_{v_j : A[\pi_i, \pi_j]>0})) \\
&= u(\boldsymbol{\Pi}\, \boldsymbol{U}_{\pi_i},\ \phi(\{m(\boldsymbol{\Pi}\, \boldsymbol{U}_{\pi_i},\ \boldsymbol{\Pi}\, \boldsymbol{U}_k,\ \boldsymbol{y}_{\pi_i k})\}_{v_k : A[\pi_i, k]>0})) && (\pi \text{ bijective}) \\
&= u(\boldsymbol{\Pi}\, \boldsymbol{U}_{\pi_i},\ \boldsymbol{\Pi}\, \phi(\{m(\boldsymbol{U}_{\pi_i}, \boldsymbol{U}_k, \boldsymbol{y}_{\pi_i k})\}_{v_k : A[\pi_i, k]>0})) && (\phi, m \text{ equivariant}) \\
&= \boldsymbol{\Pi}\, u(\boldsymbol{U}_{\pi_i}, \phi(\{m(\boldsymbol{U}_{\pi_i}, \boldsymbol{U}_k, \boldsymbol{y}_{\pi_i k})\}_{v_k : A[\pi_i, k]>0})) && (u \text{ equivariant}) \\
&= \boldsymbol{\Pi}\, f(\mathbf{U}, \mathbf{Y}, \boldsymbol{A})[\pi_i, :, :] \\
&= (\pi^{-1}.f(\mathbf{U}, \mathbf{Y}, \boldsymbol{A}))[i, :, :],
\end{aligned}
$$

which matches the definition of equivariance.

# B    Proof of Theorem 2

We first present the formal version of the theorem:

**Theorem 3** (Representation power – formal ). *Consider the class $S$ of simple graphs with $n$ nodes, diameter at most $\Delta$ and degree at most $d_{max}$. We assume that these graphs have respectively $c_X$ and $c_Y$ attributes on the nodes and the edges. There exists a permutation equivariant SMP network $f : \mathbb{R}^{n \times n} \mapsto \mathbb{R}^{n \times n \times c}$ of depth at most $\Delta + 1$ and width at most $2d_{max} + c_X + n\, c_Y$ such that, for any two graphs $G$ and $G'$ in $S$ with respective adjacency matrices, node and edge features $\boldsymbol{A}, \boldsymbol{X}, \mathbf{Y}$ and $\boldsymbol{A}', \boldsymbol{X}', \mathbf{Y}'$, the following statements hold for every $v_i \in V$ and $v_j \in V'$:*

- *If $G$ and $G'$ are not isomorphic, then for all $\pi \in \mathfrak{S}_{\mathfrak{n}}$,*

$$\boldsymbol{\Pi}^T f(\boldsymbol{A}, \boldsymbol{X}, \mathbf{Y})[i, :, :] \neq f(\boldsymbol{A}', \boldsymbol{X}', \mathbf{Y}')[j, :, :].$$

- *If $G$ and $G'$ are isomorphic, then for some $\pi \in \mathfrak{S}_n$ independent of $i$ and $j$,*

$$\boldsymbol{\Pi}^T f(\boldsymbol{A}, \boldsymbol{X}, \mathbf{Y})[i, :, :] = f(\boldsymbol{A}', \boldsymbol{X}', \mathbf{Y}')[j, :, :].$$

The fact that embeddings produced by isomorphic graphs are permutations one of another is a consequence of equivariance, so we are left to prove the first point. To do so, we will first ignore the features and prove that there is an SMP that maps the initial one-hot encoding of each node to an embedding that allows to reconstruct the adjacency matrix. The case of attributed graphs and the statement of the theorem will then follow easily.

Consider a simple connected graph $G = (V, E)$. For any layer $l \in \mathbb{N}$ and node $v_i \in V$, we denote by $G_i^{(l)} = (V, E_i^{(l)})$ the graph with node set $V$ and edge set

$$E_i^{(l)} = \{(v_p, v_q) \in E,\ d(v_i, v_p) \leq l,\ d(v_i, v_q) \leq l,\ d(v_i, v_p) + d(v_i, v_q) < 2l\}.$$

These edges correspond to the receptive field of node $v_i$ after $l$ layers of message-passing. We denote by $\boldsymbol{A}_i^{(l)}$ the adjacency matrix of $G_i^{(l)}$.

## B.1 Warm up: nodes manipulate n x n matrices

To build intuition, it is useful to first consider the case where $\boldsymbol{U}_i$ are $n \times n$ matrices (rather than $n \times c$ as in SMP). In this setting, messages are $n \times n$ matrices as well. If the initial state of each node $v_i$ is its one-hop neighbourhood ($\boldsymbol{U}_i^{(1)} = \boldsymbol{A}_i^{(1)}$), then each node can easily recover the full adjacency matrix by updating its internal state as follows:

$$\boldsymbol{U}_i^{(l+1)} = \max_{v_j \in N_i \,\cup\, v_i} \{\boldsymbol{U}_j^{(l)}\}, \tag{2}$$

where the max is taken element-wise.

**Lemma 2.** *Recursion 2 yields $\boldsymbol{U}_i^{(l)} = \boldsymbol{A}_i^{(l)}$.*

*Proof.* We prove the claim by induction. It is true by construction for $l = 1$. For the inductive step, suppose that $\boldsymbol{U}_i^{(l)} = \boldsymbol{A}_i^{(l)}$. Then,

$$\boldsymbol{U}_i^{l+1}[p,q] = 1 \iff \exists\, v_j \in \{N_i \cup v_i\} \quad \text{such that} \quad \boldsymbol{A}_j^{(l)}[p,q] = 1$$
$$\iff (v_p, v_q) \in E \text{ and } \exists\, v_j \in \{N_i \cup v_i\},\ d(v_j, v_p) \le l,\ d(v_j, v_q) \le l,\ d(v_j, v_p) + d(v_j, v_q) < 2l$$
$$\implies (v_p, v_q) \in E,\ d(v_i, v_p) \le l+1,\ d(v_i, v_q) \le l+1,\ d(v_i, v_p) + d(v_i, v_q) < 2(l+1)$$
$$\implies \boldsymbol{A}_i^{(1+1)}[p,q] = 1$$

Conversely, if $\boldsymbol{A}_i^{(l+1)}[p,q] = 1$, then there exists either a path of length $l$ of the form $(v_i, v_j, \ldots, v_p)$ or $(v_i, v_j, \ldots, v_q)$. This node $v_j$ will satisfy $\boldsymbol{U}_j^{(l)}[p,q] = 1$ and thus $\boldsymbol{U}_i^{(l+1)}[p,q] = 1$. $\quad\square$

It is an immediate consequence that, for every connected graph of diameter $\Delta$, we have $\boldsymbol{U}_i^{(\Delta)} = \boldsymbol{A}$.

## B.2 SMP: nodes manipulate node embeddings

We now shift to the case of SMP. We will start by proving that we can find an $n \times 2d_{\max}$ embedding matrix (rather than $n \times n$) that still allows to reconstruct $\boldsymbol{A}_i^{(l)}$. For this purpose, we will use the following result:

**Lemma 3** (Maehara and Rödl [46])**.** *For any simple graph $G = (V, E)$ of $n$ nodes and maximum degree $d_{max}$, there exists a unit-norm embedding of the nodes $\boldsymbol{X} \in \mathbb{R}^{n \times 2d_{max}}$ such that*

$$\forall (v_i, v_j) \in V^2,\ (v_i, v_j) \in E \iff \boldsymbol{X}_i \perp \boldsymbol{X}_j.$$

In the following we assume the perspective of some node $v_i \in V$. Let $\boldsymbol{U}_i^{(l)} \in \mathbb{R}^{n \times c_l}$ be the context of $v_i$. Further, write $\boldsymbol{u}_j^{(l)} = \boldsymbol{U}_i^{(l)}[j,:] \in \mathbb{R}^{c_l}$ to denote the embedding of $v_j$ at layer $l$ from the perspective of $v_i$. Note that, for simplicity, the index $i$ is omitted.

**Lemma 4.** *There exists a sequence $(f_l)_{l \ge 1}$ of permutation equivariant SMP layers defining $\boldsymbol{U}_i^{(l+1)} = f^{(l+1)}(\boldsymbol{U}_i^{(l)}, \{\boldsymbol{U}_j^{(l)}\}_{j \in N_i})$ such that $\boldsymbol{u}_j^{(l)} \perp \boldsymbol{u}_k^{(l)} \iff (v_j, v_k) \in E_i^{(l)}$ for every layer $l$ and nodes $v_j, v_k \in V$. These functions do not depend on the choice of $v_i \in V$.*

*Proof.* We use an inductive argument. An initialization (layer $l = 1$), we have $\boldsymbol{U}_j^{(0)} = \boldsymbol{\delta}_j$ for every $v_j$. We need to prove that there exists $\boldsymbol{U}_i^{(1)} = f^{(1)}(\boldsymbol{U}_i^{(0)}, \{\boldsymbol{U}_j^{(0)}\}_{v_j \in N_i})$ which satisfies

$$\forall (v_j, v_k) \in V^2,\ \boldsymbol{u}_j^{(1)} \perp \boldsymbol{u}_k^{(1)} \iff (v_j, v_k) \in E_i^{(1)}.$$

Rewritten in matrix form, it is sufficient to show that there exists $\boldsymbol{U}_i^{(1)}$ such that $\boldsymbol{U}_i^{(1)}(\boldsymbol{U}_i^{(1)})^\top = \mathbf{1}\mathbf{1}^\top - \boldsymbol{A}_i^{(1)}$, with $\mathbf{1}$ being the all-ones vector. $\boldsymbol{A}_i^{(1)}$ is the adjacency matrix of a star consisting of $v_i$ at the center and all its $d_i$ neighbors at the spokes. Further, it can be constructed in an equivariant manner from the layer's input as follows:

$$\boldsymbol{A}_i^{(1)} = \sum_{v_j \in N_i} \boldsymbol{\delta}_i \boldsymbol{\delta}_j^\top + \sum_{v_j \in N_i} \boldsymbol{\delta}_j \boldsymbol{\delta}_i^\top.$$

Since the rank of $\boldsymbol{A}_i^{(1)}$ is at most $d_i$ (there are $d_i$ non-zero rows), the rank of $\mathbf{1}\mathbf{1}^\top - \boldsymbol{A}_i^{(1)}$ is at most $d_i + 1 \le 2d_i \le 2d_{max}$. It directly follows that there exists a matrix $\boldsymbol{U}_i^{(1)}$ of dimension $n \times 2d_{max}$ which satisfies $\boldsymbol{U}_i^{(1)}(\boldsymbol{U}_i^{(1)})^\top = \mathbf{1}\mathbf{1}^\top - \boldsymbol{A}_i^{(1)}$. Further, as the construction of this matrix is based on the eigendecomposition of $\boldsymbol{A}_i^{(1)}$, it is permutation equivariant as desired.

*Inductive step.* According to the inductive hypothesis, we suppose that:

$$\boldsymbol{u}_j^{(l)} \perp \boldsymbol{u}_k^{(l)} \iff (v_j, v_k) \in E_i^{(l)} \text{ for all } v_j, v_k \in V$$

The function $f^{(l+1)}$ builds the embedding $\boldsymbol{U}_i^{(l+1)}$ from $(\boldsymbol{U}_i^{(l)}, \{\boldsymbol{U}_j^{(l)}, \ v_j \in N_i\})$ in three steps:

Step 1. Each node $v_j \in N_i$ sends its embedding $\boldsymbol{U}_j^{(l)}$ to node $v_i$. This is done using the message function $m^{(l)}$.

Step 2. The aggregation function $\phi$ reconstructs the adjacency matrix $\boldsymbol{A}_j^{(l)}$ of $G_j^{(l)}$ from $\boldsymbol{U}_j^{(l)}$ for each $v_j \in N_i \cup \{v_i\}$. This is done by testing orthogonality conditions, which is a permutation equivariant operation. Then, it computes $\boldsymbol{A}_i^{(l+1)}$ as in Lemma 2 using $\boldsymbol{A}_i^{(l+1)} = \max(\{\boldsymbol{A}_j^{(l+1)}\}_{v_j \in N_i \cup \{v_i\}})$, with the maximum taken entry-wise.

Step 3. The update function $u^{(l)}$ constructs an embedding matrix $\boldsymbol{U}_i^{(l+1)} \in \mathbb{R}^{n \times 2d_{max}}$ that allows to reconstruct $\boldsymbol{A}_i^{(l+1)}$ through orthogonality conditions. The existence of such an embedding is guaranteed by Lemma 1. This operation can be performed in a permutation equivariant manner by ensuring that the order of the rows of $\boldsymbol{U}_i^{(l+1)}$ is identical with that of $\boldsymbol{A}_i^{(l+1)}$.

Therefore, the constructed embedding matrix $\boldsymbol{U}_i^{(l+1)}$ satisfies

$$\boldsymbol{u}_j^{(l+1)} \perp \boldsymbol{u}_k^{(l+1)} \iff (v_j, v_k) \in E_i^{(l+1)} \text{ for all } v_j, v_k \in V$$

and the function $f^{(l+1)}$ is permutation equivariant (as a composition of equivariant functions). $\quad\square$

It is a direct corollary of Lemma 1 that, when the depth is at least as large as the graph diameter, such that $E_i^{(l)} = E$ for all $v_i$ and the width is at least as large as $2d_{\max}$, then there exist a permutation equivariant SMP $f = f^{(L)} \circ \ldots \circ f^{(1)}$ that induces an injective mapping from the adjacency matrix $\boldsymbol{A}$ to the local context $\boldsymbol{U}_i$ of each node $v_i$. As a result, given two graphs $G$ and $G'$, if there are two nodes $v_i \in V$ and $v_j' \in V'$ and a permutation $\pi \in \mathfrak{S}_n$ such that $\boldsymbol{U}_i^{(L)} = \boldsymbol{\Pi}^T \boldsymbol{U}_j'^{(L)}$, then the orthogonality conditions will yield $\boldsymbol{A} = \boldsymbol{\Pi}^T \boldsymbol{A}' \boldsymbol{\Pi}$. The contraposition is that if two nodes belong to graphs that are not isomorphic, their embedding will belong to two different equivalence classes (i.e. they will be different even up to permutations).

### B.3 Extension for attributed graphs

For attributed graphs, the reasoning is very similar: we are looking for a SMP network that maps the attributes to a set of local context matrices such that all the attributes of the graph can be recovered from the context matrix at any node. We treat the case of node and edge attributes separately:

**Node attributes**  Using $c_X$ extra channels in SMP is sufficient to create the desired embedding. We recall that the input to the SMP is a local context such that the $i$-th row of $v_i$ contains $[1, \boldsymbol{x}_i]$, where $\boldsymbol{x}_i$ is the vector of attributes of $v_i$, while the other rows are zero. Ignoring the first entry of this vector (which was used to reconstruct the topology), we propose the following update rule:

$$\boldsymbol{U}_i^{(l+1)} = \boldsymbol{U}_{j_0}^{(l)} \quad \text{where} \quad j_0 = \text{argmax}(\{|\boldsymbol{U}_j^{(l)}|\}_{j \in \{v_i \cup N_i\}}) \tag{3}$$

where the max is taken element-wise on each entry of the matrix. This function is simply an extension of the max aggregator that allows to replace the zeros of the local context by non zero values, even if they are negative. Using it, each node can progressively fill the rows corresponding to nodes that are more and more distant. With the assumption that the graph is connected, each node will eventually have access to all node features.

**Edge attributes**    As each edge attribute can be seen as a $n \times n$ matrix, edges attributes are handled in a very similar way as the adjacency matrix of unattributed graphs. If nodes could send $n \times n$ matrices as messages, they would be able to recover all the edge features using the previous update rule (equation 3). However, SMP manipulates embeddings that transform under the action of a permutation as $\pi \cdot \boldsymbol{U}_i = \boldsymbol{\Pi}^T \boldsymbol{U}_i$, whereas a $n \times n$ matrix $\boldsymbol{M}$ transforms as $\pi \cdot \boldsymbol{M} = \boldsymbol{\Pi}^T \boldsymbol{M} \boldsymbol{\Pi}$. As a result, we cannot directly pass the incomplete edge features as messages, and we need to embed them into a matrix that permutes in the right way.

The construction of a SMP that embeds the input to local contexts that allow to reconstruct an edge feature matrix $\boldsymbol{E}$ is the same as for the adjacency matrix, except for one difference: lemma 1, which was used to embed adjacency matrices into a smaller matrix cannot be used anymore, as it is specific to unweighted graphs. Therefore, we propose another way to embed each matrix $\boldsymbol{E}_i^{(l)}$ obtained at node $v_i$ after $l$ message passing layers:

- For undirected graphs, $\boldsymbol{E}_i^{(l)}$ is symmetric. We can therefore compute its eigendecomposition $\boldsymbol{E}_i^{(l)} = \boldsymbol{V} \boldsymbol{\Lambda} \boldsymbol{V}^T$.
- We add a given value $\lambda$ to the diagonal of $\boldsymbol{\Lambda}$ to make sure that all coefficients are non-negative.
- We compute the square root matrix $\boldsymbol{U} = \boldsymbol{V}(\boldsymbol{\Lambda} + \lambda \boldsymbol{I})^{1/2}$. This matrix permutes as desired under the action of a permutation: $\pi \cdot \boldsymbol{U} = \boldsymbol{\Pi}^T \boldsymbol{U}$. In addition, it allows to reconstruct the matrix $\boldsymbol{E}_i^{(l)} = \boldsymbol{U} \boldsymbol{U}^T$, so that it constitutes a valid embedding for the rest of the proof.

Note that the square root matrix permutes as desired, but that it does not compress the representation of $\boldsymbol{E}_i^{(l)}$: for each edge features, $n$ additional channels are needed, so that a SMP should have $n \times c_Y$ more channels to be able to reconstruct all edge features.

### B.4    Conclusion

We have shown that there exists an SMP that satisfies the conditions of the theorem, and specifically, we demonstrated that each layer can be decomposed in a message, aggregation and update functions that should be able to internally manipulate $n \times n$ matrices in order to produce embeddings of size $n \times 2d_{\max} + c_X + n\, c_Y$.

The main assumption of our proof is that the aggregation and update functions can *exactly* compute any function of their input — this is impossible in practice. An extension of our argument to a universal approximation statement would entail substituting the aggregation and update functions by appropriate universal approximators. In particular, the aggregation function manipulates a set of $n \times c$ matrices, which can be represented as a $n \times n \times c$ tensor with some lines zeroed out. Some universal approximators of equivariant functions for these tensors are known [48], but they have large memory requirements. Therefore, proving that a given parametrization of an SMP can be used to approximately reconstruct the adjacency matrix hinges on the identification of a simple universal approximator of equivariant functions on $n \times n \times c$ tensors.

## C    Proof of Corollary 1

Lemma 1 proves the existence of an injective mapping from adjacency matrices of simple graphs to features for a set of nodes. Therefore, any permutation equivariant function $h_{\text{eq}}(\boldsymbol{A})$ on adjacency matrices can be expressed by an equivariant function on sets

$$h_{\text{eq}}(\boldsymbol{A}) = h'_{\text{eq}}(\boldsymbol{U}) \quad \text{with} \quad \boldsymbol{U}[i,:] = \boldsymbol{u}_i \in \mathbb{R}^{2d_{\max}+c_X+n\,c_Y} \quad \forall v_i \in V,$$

as long as the node embeddings $\boldsymbol{u}_1, \dots, \boldsymbol{u}_n$ allow the reconstruction of $\boldsymbol{A}$, e.g., through orthogonality conditions. It was proven in Theorem 3 that, under the corollary's conditions, the local context $\boldsymbol{U}_i^{(L)}$ of any node $v_i$ yields an appropriate matrix $\boldsymbol{U}$. In order to compute $h_{\text{eq}}$, each node can then rely on the universal $\sigma$ to compute the invariant function:

$$h''_{\text{inv}}(\boldsymbol{U}, \mathbb{1}_i) = h'_{\text{eq}}(\boldsymbol{U})[i,:] = h_{\text{eq}}(\boldsymbol{A})[i,:] \in \mathbb{R}^c.$$

For invariant functions $h_{in}(\boldsymbol{A}) \in \mathbb{R}^c$, it suffices to build the equivariant function $h_{eq}(\boldsymbol{A}) = [h_{in}(\boldsymbol{A}), \ldots, h_{in}(\boldsymbol{A})] \in \mathbb{R}^{n \times c}$. Then, if each node $v_i$ computes $h_{\text{eq}}(\boldsymbol{A})[i,:] = h_{in}(\boldsymbol{A})$, averaging will yield $\frac{1}{n} \sum_{v_i \in V} h_{\text{eq}}(\boldsymbol{A})[i,:] = h_{in}(\boldsymbol{A})$, as required.

## D  Proof of Proposition 1

**SMPs are at least as powerful as MPNNs**  We will show by induction that any MPNN can be simulated by an SMP:

**Lemma 5.** *For any MPNN mapping initial node features $(\boldsymbol{x}_i^{(0)})_{v_i \in V}$ to $(\boldsymbol{x}_i^{(L)})_{v_i \in V}$, there is an SMP with the same number of layers such that*

$$\forall\, v_i \in V, \ \forall\, l \le L, \ \mathsf{U}^{(l)}[i,i,:] = \boldsymbol{x}_i^{(l)} \quad and \quad \forall j \neq i, \ \mathsf{U}^{(l)}[i,j,:] = 0.$$

*Proof.* Consider a graph with node features $(\boldsymbol{x}_i^{(0)})_{v_i \in V}$ and edge features $(\boldsymbol{y}_{ij})_{(v_i,v_j) \in E}$.

*Initialization*: The context tensor is initialized by mapping the node features on the diagonal of $\mathsf{U}$: $\mathsf{U}^{(0)}[i,i,:] = \boldsymbol{x}_i^{(0)}$. The desired property is then true by construction.

*Inductive step*: Denote by $(\boldsymbol{x}_i^{(l)})_{v_i \in V}$ the features obtained after $l$ layers of the MPNN. Assume that there is a k-layer SMP such that the local context after $l$ layers contains the same features in its diagonal elements: $\mathsf{U}^{(l)}[i,i,:] = \boldsymbol{x}_i^{(l)}$ and 0 in the other entries. Consider one additional layer of MPNN:

$$\boldsymbol{x}_i^{(l+1)} = u(\boldsymbol{x}_i^{(l)}, \phi(\{m(\boldsymbol{x}_i^{(l)}, \boldsymbol{x}_j^{(l)}, \boldsymbol{y}_{ij})\}_{j \in N_i}))$$

and the following SMP layer:

$$\boldsymbol{U}_i^{(l+1)} = diag(\tilde{u}(\boldsymbol{U}_i^{(l)}, \tilde{\phi}(\{\tilde{m}(\mathbf{11}^T \boldsymbol{U}_i^{(l)}, \mathbf{11}^T \boldsymbol{U}_j^{(l)}, \boldsymbol{y}_{ij})\}_{j \in N_i}))),$$

where $\tilde{m}$, $\tilde{\phi}$ and $\tilde{u}$ respectively apply the functions $m, \phi$ and $u$ simultaneously on each line of the local context $\boldsymbol{U}_i$. As the only non-zero line of $\boldsymbol{U}_i$ is $\boldsymbol{U}_i[i,:]$, $\mathbf{11}^T\boldsymbol{U}_i^{(l)}$ replicates the $i$-th line of $\boldsymbol{U}_i^{(l)}$ on all the other lines, so that they all share the same content $\boldsymbol{x}_i^{(l)}$. After the application of the message passing functions $\tilde{m}, \tilde{\phi}$ and $\tilde{u}$, all the lines of $\boldsymbol{U}_i$ therefore contain $\boldsymbol{x}_i^{(l+1)}$.

Finally, the function *diag* extracts the main diagonal of the tensor $\mathsf{U}$ along the two first axes. Let $\delta_{i,j}$ be the function that is equal to 1 if $i = j$ and 0, otherwise. We have: $diag(\mathsf{U})[i,j,:] = \mathsf{U}[i,j,:] \, \delta_{i,j}$. Note that this function can equivalently be written as an update function applied separately to each node: $diag(\boldsymbol{U}_i)[j,:] = \boldsymbol{U}_i[j,:]\delta_{i,j}$. We now have $\mathsf{U}^{(l+1)}[i,i;:] = \boldsymbol{x}_i^{l+1}$ and $\mathsf{U}$ equal to 0 on all the other entries, so that the induction hypothesis is verified at layer $l+1$. $\qquad\square$

As any MPNN can be computed by an SMP, we conclude that SMPs are at least as powerful as MPNNs.

**SMP are strictly more powerful**  To prove that SMPs are strictly more powerful than MPNNs, we use a similar argument to [22, 32]:

**Lemma 6.** *There is an SMP network which yields different outputs for the two graphs of Fig. 3, while any MPNN will view these graphs are isomorphic.*

Figure 3: While MPNNs cannot distinguish between two regular graphs such as these ones, SMPs can.

*Proof.* The two graphs of Fig. 3 are regular, which implies that they cannot be distinguished by the Weisfeiler-Lehman test or by MPNNs without special node features [32]. On the contrary, consider an SMP $f$ made of three layers computing $\boldsymbol{U}_i^{(l+1)} = \sum_{v_i \in N_i} \boldsymbol{U}_i^{(l)}$, followed by the trace of $\boldsymbol{U}^{(3)}$ as a a pooling function. As each layer can be written $\boldsymbol{U}^{(l+1)} = \boldsymbol{A}\boldsymbol{U}^{(l)}$ and $\boldsymbol{U}^{(0)} = \boldsymbol{I}_n$, we have $f(\boldsymbol{A}) = tr(\boldsymbol{A}^3)$. In particular $f(\boldsymbol{A}) = 2$ for the graph on the left, while $f(\boldsymbol{A}) = 0$ on the right. $\square$

## E   A more compact representation with graph coloring

In SMP, the initial local context is a one-hot encoding of each node: $\boldsymbol{U}_i^{(0)} = \boldsymbol{\delta}_i \in \mathbb{R}^n$. When the graph diameter $\Delta$ is large compared to the number of layers $L$, the memory requirements of this one-hot encoding can be reduced by attributing the same identifiers to nodes that are far away from each other. In particular, no node should see twice the same identifier in its $L$-hop neighborhood. To do so, we propose to build a graph $G'$ where all $2L$-hop neighbors of $G$ are connected, and to perform a greedy coloring of $G'$ (Algorithm 1). Although the number of colors $\chi$ used by the greedy coloring might not be optimal, this procedure guarantees that identifiers do not conflict.

---

**Algorithm 1:** Node coloring

---

**Input:** A graph $G = (V, E)$ with $n$ nodes, $L \in \mathbb{N}$ (number of layers.)
**Output:** A binary matrix $\boldsymbol{U}_i^0 \in \mathbb{R}^{n \times \chi}$, where $\chi$ is the number of colors.
Create the graph $G' = (V, \{(i,j), d(i,j)\} \leq 2L)$
$\boldsymbol{c} \in \mathbb{R}^n \leftarrow greedy\_coloring(G')$
**return** *one_hot_encoding*$(\boldsymbol{c})$

---

The one-hot encoding of the colors $\boldsymbol{U}_i^0 \in \mathbb{R}^\chi$ is then used to initialize the local context of $v_i$. The only change in the SMP network is that in order to update the representation that node $i$ has of node $j$, we now update $\boldsymbol{U}_i[c_j, :]$ instead of $\boldsymbol{U}_i[j, :]$, where $c_j$ is the color associated to node $v_j$. Note however that the coloring is only useful if the graph has a diameter $\Delta > 2L$. This is usually the case in geometric graphs such as meshes, but often not in scale-free networks.

## F   Proof of Proposition 2

We will prove by induction that any Fast SMP layer can be approximated by two blocks of PPGN. It implies that the expressive power of Fast SMP is bounded by that of PPGN.

Recall that a block of PPGN is parameterized as:

$$\mathbf{T}^{(l+1)} = m_4(m_3(\mathbf{T}^{(l)}) \| m_1(\mathbf{T}^{(l)}) @ m_2(\mathbf{T}^{(l)})),$$

where $m_k$ are MLPs acting over the third dimension of $\mathbf{T} \in \mathbb{R}^{n \times n \times c}$: $\forall (i,j), \ m_k(\mathbf{T})[i,j,:] = m_k(\mathbf{T}[i,j,:])$. Symbol $\|$ denotes concatenation along the third axis and @ matrix multiplication performed in parallel on each channel: $(\mathbf{T} @ \mathbf{T}')[:,:,c] = \mathbf{T}[:,:,c] \ \mathbf{T}'[:,:,c]$.

To simplify the presentation, we assume that:

- At each layer $l$, one of the channels of $\mathbf{T}^{(l)}$ corresponds to the adjacency matrix $\boldsymbol{A}$, another contains a matrix full of ones $\mathbf{1}_n \mathbf{1}_n^\top$ and a third the identity matrix $\boldsymbol{I}_n$, so that each PPGN layer has access at all times to these quantities. These matrices can be computed by the first PPGN layer and then kept throughout the computations using residual connections.
- The neural network can compute entry-wise multiplications $\odot$. This computation is not possible in the original model, but it can be approximated by a neural network.
- $\mathbf{U}$ and $\mathbf{T}$ have only one channel (so that we write them $\boldsymbol{U}$ and $\boldsymbol{T}$). This hypothesis is not necessary, but it will allow us to manipulate matrices instead of tensors.

**Initialization**   Initially, we simply use the same input for PPGN as for SMP ($\boldsymbol{U}^{(0)} = \boldsymbol{T}^{(0)} = \boldsymbol{I}_n$).

**Induction** Assume that at layer $l$ we have $\boldsymbol{U}^{(l)} = \boldsymbol{T}^{(l)}$. Consider a layer of Fast SMP:

$$\boldsymbol{U}_i^{(l+1)} = \frac{1}{d_{\text{avg}}} \left( \sum_{v_j \in N_i} \hat{\boldsymbol{U}}_j^{(l)} + \hat{\boldsymbol{U}}_i^{(l)} \boldsymbol{W}_4^{(l)} \odot \sum_{v_j \in N_i} \hat{\boldsymbol{U}}_j^{(l)} \boldsymbol{W}_5^{(l)} \right),$$

where

$$\hat{\boldsymbol{U}}_i^{(l)} = \boldsymbol{U}_i^{(l)} \boldsymbol{W}_1^{(l)} + \frac{1}{n} \mathbf{1}_n \mathbf{1}_n^T \boldsymbol{U}_i^{(l)} \boldsymbol{W}_2^{(l)} + \mathbf{1}_n (\boldsymbol{c}^{(l)})^\top + \frac{1}{n} \mathbb{1}_i \mathbf{1}^T \boldsymbol{U}_i^{(l)} \boldsymbol{W}_3^{(l)}.$$

A first PPGN block can be used to compute $\hat{\boldsymbol{U}}_i^{(l)}$ for each node. This block is parametrized by:

$$m_1(\boldsymbol{U}^{(l)}) = \frac{1}{n} \mathbf{1_n} \mathbf{1}_n^T, \qquad\qquad m_2(\boldsymbol{U}^{(l)}) = \boldsymbol{U}^{(l)},$$

$$m_3(\boldsymbol{U}^{(l)}) = \boldsymbol{U}^{(l)} \boldsymbol{W}_1 + \mathbf{1}\boldsymbol{c}^T + (\boldsymbol{I}_n \odot \boldsymbol{U}^{(l)}) \boldsymbol{W}_3, \qquad m_4(\hat{\boldsymbol{U}}, \tilde{\boldsymbol{U}}) = \hat{\boldsymbol{U}} + \tilde{\boldsymbol{U}} \boldsymbol{W}_2 + \boldsymbol{I}_n \odot (\tilde{\boldsymbol{U}} \boldsymbol{W}_3)$$

The output of this block exactly corresponds to $\hat{\boldsymbol{U}}^{(l)}$. Then, a second PPGN block can be used to compute the rest of the Fast SMP layer. It should be parametrized as:

$$m_1([\hat{\boldsymbol{U}}^{(l)}]) = \boldsymbol{A} \, / \, \bar{d}, \qquad\qquad m_2([\hat{\boldsymbol{U}}^{(l)}]) = \hat{\boldsymbol{U}}^{(l)},$$

$$m_3([\hat{\boldsymbol{U}}^{(l)}]) = \hat{\boldsymbol{U}}^{(l)}, \qquad\qquad m_4(\hat{\boldsymbol{U}}^{(l)}, \tilde{\boldsymbol{U}}) = \tilde{\boldsymbol{U}} + (\hat{\boldsymbol{U}}^{(l)} \boldsymbol{W_4}) \odot (\tilde{\boldsymbol{U}} \boldsymbol{W_5})$$

By plugging these expressions into the definition of a PPGN block, we obtain that the output of this block corresponds to $\boldsymbol{U}^{(l+1)}$ as desired.

# G Comparison between SMP, Provably powerful graph networks and Ring-GNN

Figure 4: Training curves of SMP, PPGN and Ring-GNN for different cycle lengths $k$. NLL stands for negative log-likelihood. Red dots indicate the epoch when SMP training was stopped. The training loss sometimes exhibits peaks of very high value which last one epoch – they were removed for readability. Provably powerful graph networks are much more difficult to train than SMP: their failure is not due to a poor generalization, but to the difficulty of optimizing them. Ring-GNN works well for small graphs, but we did not manage to train it with the largest graphs (66 or 72 nodes). We attribute this phenomenon to an inductive bias that is less suited to the task. PPGN and SMP training time per epoch are approximately the same, while RING-GNN is between two and three times slower.