[Reviews · NeurIPS 2020]

Review 1

Summary and Contributions: ------------------------------------------------------------------- Post Rebuttal: Thank you authors for the thoughtful responses to the reviews. After reading the response from the authors and the ensuing discussions - I realized the model was indeed inductive and hence increased my score by 1 (from 4 to 5). However, in discussions with other reviewers, we also discussed a simple counter example which shows that local isomorphisms are not always preserved with the proposed model - and hence requires comprehensive comparison with powerful equivariant models like PGNN, SEAL, etc for the table 3. ------------------------------------------------------------------- This work proposes SMP [structural message passing] graph neural network which propagates matrices among neighboring nodes in a graph rather than vectors as messages - and provides theoretical guarantees about its expressiveness in comparison to Message Passing Neural Networks (MPNNs) based on the 1-WL (Weisfeiler Leman) algorithm. Since a direct implementation of the model could prove to be expensive, the work proposes a computationally faster variant which preserves some properties of the original model. The work validates its theoretical claims (for connected graphs) of the proposed model on tasks, with better accuracy/ lower errors in comparison to 1-WL based GNN's on both the default implementations and the faster variant.

Strengths: Representation learning for graphs has been rapidly evolving over the past few years. A vast majority of the recent works have focused on new variants of message passing neural networks, GNN's which propagate information between nodes in the form of vectors. In contrast, this work advocates for propagating information in the form of matrices (where each vertex has information regarding every other node in the graph from its viewpoint). The authors theoretically (for connected graphs) show that propagating matrices of the above form, while still preserving equivariance results in more powerful models in comparsion to 1-WL GNNs - and also provide a computationally faster variant of the originally proposed more expensive model.

Weaknesses: 1. The proposed model - uses unique identifiers associated with each node (and across epochs, nodes are not assigned different id's) and hence learns positional embeddings (G-equivariant) (like [1] which instead employs anchor points, and hence is computationally less expensive than the proposed model, and [2] - add these positional embedding strategies as baselines) rather than structural representations [3] (G-invariant) and hence is unable to obtain the same representation (as a stand alone) for isomorphic node sets [subgraphs and even isomorphic nodes] within a given graph. This model (stand alone) can inherently lead to weaker results on the vertex classification tasks(and also subgraph classification tasks - on the other hand a traditional 1-WL GNN which learns invariant representations of nodes, which are then subsequently pooled to obtain subgraph representations will obtain the same representation for isomorphic subgraphs - although they may suffer from false positives) Please highlight this and add these numbers as well. 2. Why aren't [4], [7], [8] among the baselines evaluated? The experimental tasks eg. cycle detection, geodesic distances in this work are known well to be higher order tasks - unable to be solved by 1-WL GNN's [7] and require structural representations of node sets [3] rather than individual nodes. Being a positional embedding strategy, the method compares with GNN baselines like GIN and GAT rather than comparisons with other explicit positional embedding strategies like [1], [2], etc or strategies which learn structural representations of node sets like higher order k-GNN's based on k-WL , local delta -k-WL [5], [6]. Please add these baselines - [4], [8], [7] as well as [1], [2], [5], [6] in the comparison. 3. Moreover, while theoretically possible to identify isomorphic graphs with this model, stand alone this strategy still requires to check O(n!) permutations - which is computationally expensive rather than a comparison in expectation like in [4]. 4. The proposed model is inherently transductive and is directly not applicable to larger size graphs than it is trained on. References 1. You, Jiaxuan, Rex Ying, and Jure Leskovec. "Position-aware graph neural networks." arXiv preprint arXiv:1906.04817 (2019). 2. Zhang, Muhan, and Yixin Chen. "Link prediction based on graph neural networks." Advances in Neural Information Processing Systems. 2018. 3. Srinivasan, Balasubramaniam, and Bruno Ribeiro. "On the Equivalence between Positional Node Embeddings and Structural Graph Representations." International Conference on Learning Representations. 2020. 4. Murphy, Ryan L., et al. "Relational pooling for graph representations." arXiv preprint arXiv:1903.02541 (2019). 5. Morris, Christopher, et al. "Weisfeiler and leman go neural: Higher-order graph neural networks." Proceedings of the AAAI Conference on Artificial Intelligence. Vol. 33. 2019 6. Morris, Christopher, and Petra Mutzel. "Towards a practical $ k $-dimensional Weisfeiler-Leman algorithm." arXiv preprint arXiv:1904.01543 (2019). 7. Chen, Zhengdao, et al. "Can graph neural networks count substructures?." arXiv preprint arXiv:2002.04025 (2020). 8. Zhengdao Chen, Soledad Villar, Lei Chen, and Joan Bruna. On the equivalence between graph isomorphism testing and function approximation with gnns. In Advances in Neural Information Processing Systems, pages 15868–15876, 2019.

Correctness: In proposition 1 - Please add that SMP is strictly more powerful than 1-WL based MPNN's.

Clarity: Yes, the content in the paper is well written.

Relation to Prior Work: The work has done well in capturing majority of the related work in this fast paced field. However, the work has still missed some more recent works in this field as mentioned in the weaknesses section- which highlighted the limitations of the compared baselines.

Reproducibility: Yes

Additional Feedback: Please address the weaknesses to the best possible. If adequately addressed or highlighted as limitations in the paper, I will be happy to update my scores. Other Minor Concerns: 1. In theorem 2 - maybe the first bullet point should read "for any \pi in the finite symmetric group"?


Review 2

Summary and Contributions: This paper proposes a new type of message passing graph neural network that aims to use node identifiers in an equivariant manner, i.e. guaranteeing that any permutation of the node ids would result in a correspondingly permuted representation of the graph. The main idea is to keep at each node a matrix of the full set of node embeddings and to use message passing to update this matrix based on the neighbor nodes' own representations. The authors prove that, in the case of simple graphs, if the message passing update functions are equivariant, then the entire model is equivariant and can distinguish between non-isomorphic graphs and is strictly more powerful than classical MPNNs. On two sets of experiments on synthetic graphs, the authors show that their proposed SMP model outperforms popular MPNN models on detecting cycles and other similar graph properties for which exact polynomial algorithms exist.

Strengths: See above. This work is an important contribution towards improving classic MPNN architectures by considering node IDs in an equivariant manner, even if at an added computational cost. Previous work has either used only node and edge features, in which case the respective models cannot recognize simple substructures such as cycles, or has used node IDs to tackle the weaknesses of MPNNs, but only in a permutation sensitive way.

Weaknesses: - The theory (thm 2, cor 1) on the representational power of SMPs is only for simple unlabeled graphs. Is there any similar result for graphs with node and/or edge features ? - The experiments are quite limited. I wish to have seen SMPs in the context of graphs with node and edge features and on standard benchmarks used by GCNs/GIN/GAT models. - how good is SMP at counting cycles ? - is fast SMP less expressive than SMP ? I wish to have seen more discussion on the power of different architectures. - are, in the limit of using sufficiently many layers, all embeddings of node j at each node i becoming equal ? Can this be formally proved ? If this is not true, then isn't it a weakness that each node can learn a potentially different representation of all the nodes in the graph ? - how well is fast SMP perming on the cycle detection task ? ====================== Later edit: I agree with other reviewers' comments on the lack of powerful equivariant baselines and I do believe that the current experimental setup is limited. I am lowering my score as a consequence.

Correctness: See above.

Clarity: Paper is generally well written. However, the authors should make it clear from the beginning that the theory applies only to simple unlabelled graphs.

Relation to Prior Work: Related work looks good.

Reproducibility: Yes

Additional Feedback:


Review 3

Summary and Contributions: The paper proposes a generalization of message-passing neutral networks (MPNNs), referred to as SMP, that can efficiently detect graph properties. The key idea is to include the unique node identifiers in the propagated message so that graph topology can be reconstructed. A formal analysis is given to reason about the permutation equivariant property and its representation power, which is universal in the limited and at least more powerful than MPNNs without special node features. The effectiveness of SMP is validated on the tasks requiring detection of graph properties and SMP clearly outperforms the state-of-the-art.

Strengths: Overall, this a nice paper with strong motivation and solid theoretical analysis. It broadens the capability of MPNNs by a simple idea, which, to the best of my knowledge, was not proposed and not formally analyzed by any existing variants of MPNNs or GNNs with identifiers.

Weaknesses: The proposed method increases the time/memory complexity by a factor of n (i.e., the # of nodes) compared to MPNNs. Although the authors mention a way to reduce the computational complexity using graph coloring, the applicability is still severely limited by the graph size if the graph does not have a small chromatic number.

Correctness: The analysis of the permutation equivariant property (Theorem 1) and SMP more powerful than MPNN (Proposition 1) are correct. For Theorem 2, which analyzes the best possible expressive power of SMP, I only skimmed through the proof and it seems to be correct.

Clarity: This paper is well-written. The sections of introduction and related work are well-organized to provide enough context to know the novelty of this work without assuming much background knowledge in the field of equivariant neural networks. Moreover, the main theorems are accompanied by proof sketches, interpretations, and limitations that help the reader grips an overview of their results without knowing the details.

Relation to Prior Work: Yes, the related works are thoroughly surveyed and concisely compared in terms of the technical methods and computational efficiency.

Reproducibility: Yes

Additional Feedback: Comments: * Theorem 2 states the expressive power of SMP in the limit. It would be excellent if a non-trivial lower-bound on the computational complexity for any message-passing architecture to achieve the universal approximability can be given. Suggestions / Minor points: * In Table 3, the (Fast) MPNN is for the ablation study mentioned in (line 278), right? It is more clear if it is described in the caption of Table 3. * It would be nice to include a reference to the greedy coloring algorithm in Appendix E, and briefly mention the computational requirement. * The function d(.,.) in (line 438) is not defined. I suppose it is the length of the shortest path (on unweighted graph) between the two input nodes. * About the Reference section, it might be better to use the citation referring to the published proceedings rather than the arXiv version. Typos: * (line 62) ... [1], MPPNs have ... -> "MPPNs" should be "MPNNs"? * (line 235) ... In implies ... -> "In" should be "It" * (line 297) the citation is not correctly displayed * (line 496) we demonstrated the each ... -> "the" should be "that" ------------------------------ Thank authors for the reply on the lower bound. During the discussion with other reviewers, I agree that more experimental results should be added. Especially, to compare with positional embedding methods such as P-GNN and SEAL since the proposed method falls in the category. Please add the experimental comparison or explain the reason for not doing so in the future version of the paper.


Review 4

Summary and Contributions: This submission proposes a simple way to modify message-passing graph neural networks to accommodate node identifiers in a manner equivariant to node permutation. The proposed method is analysed theoretically and implemented practically. The analytic results show that, in return for increased memory usage and computational complexity, the proposed method is more expressive than standard message-passing graph neural nets, and is capable of learning different feature maps for any pair of non-isomorphic graphs. The practical implementation is evaluated on problems related to the detection of graph properties, such as cycle detection. The paper is well-written and thorough, and the empirical results suggest that the proposed method merits further investigation. Good paper, accept.

Strengths: The method appears to be well-motivated, since it combines the strengths of message-passing graph nets (specifically ease of training) with the greater representation power characteristic of methods like provably powerful graph nets. The empirical results for learning graph properties demonstrate a compelling increase in performance, so it seems likely that this submission would be relevant to the NeurIPS research community.

Weaknesses: It appears that the proposed method would be unsuitable for some types of large graphs that commonly appear in practice, such as social network graphs, since the diameter of such graphs tends to be small. While the authors make a preliminary attempt to mitigate the memory cost of their method with a compact representation based on graph coloring, such a modification only helps if (as the authors note) the number of message-passing layers is significantly smaller than the graph diameter. It is certainly possible that architectures with representation power such as that identified in Thm 2 do not admit compressed node features for general graphs. A negative analytic result to that effect could be helpful to understand the minimum computational resources required for expressive graph nets. The practical significance of the work is somewhat difficult to evaluate since the empirical evaluation is restricted to a couple benchmarks.

Correctness: The claims and empirical methodology of this submission appear to be correct.

Clarity: The paper is well-written and thorough. My immediate questions and concerns were anticipated and clarified.

Relation to Prior Work: The submission clearly differentiates itself from prior work.

Reproducibility: Yes

Additional Feedback:

[Author Response · NeurIPS 2020]

We would like to thank the reviewers for their thorough evaluation and constructive feedback. They have been really helpful in improving our work. Below, we address the main comments. Our revisions will be incorporated in the camera-ready version along with the additional related work, corrections and comments brought forth by the reviewers.

**Implications of equivariance and independence to the initial choice of identifiers (R1)** SMP's equivariance properties ensure that a change in the one-hot encoding results in a permutation of the rows of each local context $U_i$. However, in order to produce a final output (for node or graph classification), the rows of each $U_i$ are pooled into a vector (in an equivariant manner as well). As a result, *the output of each node is independent of the initial one-hot encoding*, and the latter need not be consistent across examples and/or layers. Equivariance has some other useful consequences:

*Local isomorphisms:* if the subgraphs $G_i^k$ and $G_j^k$ induced by $G$ on the k-hop neighborhoods of $v_i$ and $v_j$ are isomorphic, then on node classification, any $k$-layer SMP $f$ will yield the same result for $v_i$ and $v_j$. To prove it, we first observe that $f(G)_{v_i} = f(G_i^k)_{v_i}$ and $f(G)_{v_j} = f(G_j^k)_{v_j}$, and then write the definition of equivariance for an isomorphism $\pi$ mapping $G_i^k$ to $G_j^k$ and $i$ to $j$.

*Transductivity:* since all equivariant functions can take a variable number of inputs, SMP can be used transductively. If a node is added to a graph, a new line is simply created in each local context and there is no need to retrain the model. Experimentally, we observed that SMP managed to generalize to larger graphs than those seen during training: e.g., when we trained SMP to detect 4-cycles on graphs with 20 nodes (where it reaches 100% test accuracy), we obtained 99.05% accuracy when evaluating the same task on graphs with 36 nodes.

*Isomorphism testing:* To test isomorphism using SMP, one can pool after each layer the $n \times d$ local context of each node into a feature vector in $\mathbb{R}^d$. For this purpose, a universal approximator of functions on sets (such as Deep Sets) can be used. Similarly to MPNNs, isomorphism can then be tested by comparing the multisets of node features after each layer. Equivariance guarantees that these multisets do not depend on the initial choice of the one-hot encoding, so that there is no need to sum over permutations—a key difference between SMP and relational pooling methods from the literature.

**Discussion about the theoretical results (R2)** We would like to elaborate on three of the reviewer's points:

*Universality with features:* Theorem 2 can be extended to attributed undirected graphs, but $d_{nodes} + nd_{edges}$ more channels are required in this case. For the node features, $d_{nodes}$ channels can be used to store the features of all nodes using a variation of max pooling. For edge features, the same sketch of proof as Theorem 2 can be used: if each node can store tensors of size $n \times n \times d_{edges}$, they can all recover the edge features. However, another embedding is needed as Lemma 1 does not apply anymore. If the graph is undirected, the square root matrix of each feature (which may be complex-valued) constitutes a valid embedding, as it permutes as desired. However, this embedding does not compress the representation, so that $n \times d_{edges}$ new channels are required. Corollary 1 follows in the same way as previously.

*Expressivity:* We do not yet have any formal results stating whether SMP is strictly more expressive than Fast SMP. Still, we observe that the proof that PPGN is at least as expressive as Fast SMP does not apply to SMP. This stems from the fact that SMP computes messages of the form $m(U_i, U_j, e_{ij})$, while PPGN can only store messages of the form $m(U_j)$.

*Equality of the embeddings in the limit:* This is an interesting question. As Lemma 1 is not constructive, it is indeed unclear at this point whether all node embeddings will become equal at infinite depth. At this point we can only observe that it is a possible scenario.

**Scalability and lower bound on the complexity (R3, R4)** Although SMP is more efficient than previous powerful equivariant methods, large graphs exhibiting the small-world property indeed constitute a challenge. In this case, the scalability of SMP can be improved by simply using fewer identifiers (and ignoring conflicts), at the cost of breaking the theoretical guarantees of the network. We plan to investigate this extension in our future work.

We also agree that lower bounds on the complexity required for universality would be very valuable to the community. We are aware of two results towards this direction: (*i*) if the feature space is continuous, at least $d_{max}$ width is required to make the aggregation function injective [20]. (*ii*) for all message-passing methods (including SMP), solving some simple combinatorial tasks necessitates depth $\times$ width $= \Omega(n)$ [18].

**Additional experiments (R1, R2, R4)** Following the reviewers' suggestion, we ran experiments on Ring-GNN and Relational Pooling (RP): (*i*) Ring-GNN could solve the cycle-detection problem up to $k = 8, n = 50$. However, in this configuration ($k = 8, n = 50$), it required $5\times$ more epochs and $10\times$ more time than SMP to converge (*ii*) RP with $\pi$-SGD (summing over 8 permutations) obtained $100\%$ accuracy on all training sets, but exhibited overfitting (which was not observed on equivariant methods): for $k = 6$, $n = 56$, test accuracy was $84.1\%$ for RP against $99.8\%$ for SMP.

Finally, we acknowledge the importance of additional benchmarking on tasks where both features and structure play a role. We are currently working on the QM9 and ZINC datasets, and plan to make the method available in Pytorch-geometric.

[Meta-Review · NeurIPS 2020]

All reviewers appreciated the proposed new type of message passing graph neural network that aims to use node identifiers in an equivariant manner, i.e. guaranteeing that any permutation of the node ids would result in a correspondingly permuted representation of the graph. They also note the increase in computation. Overall, very relevant work for the community.